# A 12-hospital prospective evaluation of a clinical decision support prognostic algorithm based on logistic regression as a form of machine learning to facilitate decision making for patients with suspected COVID-19

**Monica I. Lupei** [1,¤]*, **Danni Li**[2], **Nicholas E. Ingraham**[3], **Karyn D. Baum**[4], **Bradley Benson**[4], **Michael Puskarich**[5], **David Milbrandt**[5], **Genevieve B. Melton**[6,7], **Daren Scheppmann**[7], **Michael G. Usher**[4‡], **Christopher J. Tignanelli**[6,7,8‡]

1 Division of Critical Care, Department of Anesthesiology, University of Minnesota Medical School, Minneapolis, Minnesota, United States of America, 2 Department of Laboratory Medicine and Pathology, University of Minnesota Medical School, Minneapolis, Minnesota, United States of America, 3 Division of Pulmonary and Critical Care, Department of Medicine, University of Minnesota Medical School, Minneapolis, Minnesota, United States of America, 4 Division of General Internal Medicine, Department of Medicine, Section of Hospital Medicine, University of Minnesota Medical School, Minneapolis, Minnesota, United States of America, 5 Department of Emergency Medicine, University of Minnesota Medical School, Minneapolis, Minnesota, United States of America, 6 Department of Surgery, University of Minnesota Medical School, Minneapolis, Minnesota, United States of America, 7 Institute for Health Informatics, University of Minnesota, Minneapolis, Minnesota, United States of America, 8 Division of Critical Care and Acute Care Surgery, Department of Surgery, University of Minnesota Medical School, Minneapolis, Minnesota, United States of America

☯ These authors contributed equally to this work.
¤ Current address: Department of Anesthesiology, University of Minnesota Medical School, Minneapolis, Minnesota, United States of America
‡ MGU and CJT also contributed equally to this work
* lupei001@umn.edu

**Data Availability Statement:** Data cannot be shared publicly because of protected health

## Abstract

### Objective

To prospectively evaluate a logistic regression-based machine learning (ML) prognostic algorithm implemented in real-time as a clinical decision support (CDS) system for symptomatic persons under investigation (PUI) for Coronavirus disease 2019 (COVID-19) in the emergency department (ED).

### Methods

We developed in a 12-hospital system a model using training and validation followed by a real-time assessment. The LASSO guided feature selection included demographics, comorbidities, home medications, vital signs. We constructed a logistic regression-based ML algorithm to predict "severe" COVID-19, defined as patients requiring intensive care unit (ICU) admission, invasive mechanical ventilation, or died in or out-of-hospital. Training data included 1,469 adult patients who tested positive for Severe Acute Respiratory Syndrome

information. Data are available from the M Health Fairview Institutional Data Access / Ethics Committee (contact via Institutional Review Board, email irb@umn.edu and phone +1 (612) 626-5654) for researchers who meet the criteria for access to confidential data.

**Funding:** CJT: Funding received through Agency for Healthcare Research and the Gates Foundation KDB: Supported on 2 grants from the MN Department of Health to oversee statewide placement of patients as part of COVID response. The funders had no role in study design, data collection and analysis, decision to publish, or preparation of the manuscript.

**Competing interests:** The authors have declared that no competing interests exist.

Coronavirus 2 (SARS-CoV-2) within 14 days of acute care. We performed: 1) temporal validation in 414 SARS-CoV-2 positive patients, 2) validation in a PUI set of 13,271 patients with symptomatic SARS-CoV-2 test during an acute care visit, and 3) real-time validation in 2,174 ED patients with PUI test or positive SARS-CoV-2 result. Subgroup analysis was conducted across race and gender to ensure equity in performance.

## Results

The algorithm performed well on pre-implementation validations for predicting COVID-19 severity: 1) the temporal validation had an area under the receiver operating characteristic (AUROC) of 0.87 (95%-CI: 0.83, 0.91); 2) validation in the PUI population had an AUROC of 0.82 (95%-CI: 0.81, 0.83). The ED CDS system performed well in real-time with an AUROC of 0.85 (95%-CI, 0.83, 0.87). Zero patients in the lowest quintile developed "severe" COVID-19. Patients in the highest quintile developed "severe" COVID-19 in 33.2% of cases. The models performed without significant differences between genders and among race/ethnicities (all p-values > 0.05).

## Conclusion

A logistic regression model-based ML-enabled CDS can be developed, validated, and implemented with high performance across multiple hospitals while being equitable and maintaining performance in real-time validation.

## Introduction

The dynamic of Severe Acute Respiratory Syndrome Coronavirus 2 (SARS-CoV-2) infection raised concerns regarding resource availability throughout medical systems, including intensive care unit (ICU) healthcare providers, personal protective equipment, total hospital, and ICU beds, and mechanical ventilators. On March 11[th], 2020, the World Health Organization declared the Coronavirus disease 2019 (COVID-19) a pandemic. The COVID-19 pandemic has caused over 249 million confirmed infections and over 5 million confirmed deaths as of November 9[th], 2021 [1]. One of the initial large observational studies, published from China, revealed that approximately 15% of the confirmed cases required hospitalization, 5% needed ICU admission, and 2.3% died [2]. A multihospital United States (U.S.) based cohort study identified that the 30-day mean risk standardized event rate of hospital mortality and hospice referral among patients with COVID-19 varied from 9% to 16%, with better outcomes occurring in community's with lower disease prevalence [3]. A large cross-sectional study found racial and ethnic disparities in rates of COVID-19 hospital and ICU admission and in-hospital mortality in the US [4].

Since the beginning, global efforts by the scientific community to understand SARS-CoV-2 and the COVID-19 from the bench to the bedside have been remarkable [5]. Stratifying disease severity is an essential aspect of patient care; however, during a pandemic, its role becomes paramount and expands to improving patient safety while also optimizing hospital resource utilization. Several studies have developed emergency department (ED) evaluation systems with variable goals and methods [6–12]. These models successfully evaluated the possibility of isolating COVID-19 patients in ED, the epidemiology and COVID-19 clinical data, the

advantage of distinguishing life-threatening emergencies, and the likelihood of COVID-19 diagnosis [6–12].

Most predictive models for COVID-19 severity involved patients with a positive polymerase chain reaction (PCR) test, not in patients with suspected COVID-19. A systematic evaluation of COVID-19 predictive models aimed at identifying clinical deterioration found that the majority of published studies included patients with confirmed infection [13], making them less useful in the clinic or emergency departments when diagnosis remains uncertain. The majority of predictive models for patients with suspected COVID-19 infection aimed to diagnose COVID-19, and very few predicted severity [14]. One systematic review of the prognostic models emphasized the high risk of bias while not recommending their use in clinical practice yet [15]. Since limitations mark the systematic reviews of the prognostic models, and a group of researchers from the United Kingdom (UK) developed a COVID-19 precise living document [16]. Another group of researchers proposed an open platform for such reviews that will be continuously updated using artificial intelligence and numerous experts [17]. The QCOVID is a published living risk prediction algorithm that performed well for predicting time to death in patients with confirmed or suspected COVID-19 [18].

We hypothesize that a logistic regression-based machine learning (ML) tool for patients with suspected or confirmed COVID-19 can accurately and equitably predict the development of "severe" COVID-19. The objective of this study was to conduct a 12-site prospective observational study to evaluate the real-time performance of a ML-enabled COVID-19 prognostic tool delivered as clinical decision support (CDS) to ED providers to facilitate shared decision-making with patients regarding ED discharge.

## Methods

### Study design and setting

This is a retrospective and prospective multihospital observational study that developed, implemented, and evaluated a prognostic model in patients with PCR-confirmed COVID-19 diagnosis or suspected COVID-19 (person under investigation [PUI]) in a 12-hospital system. This study was approved and determined as non-human research by the University of Minnesota Institutional Review Board (STUDY00011742).

### Selection of participants

Patients were included if they were PCR confirmed COVID-19 positive or symptomatic PUI with a patient status of emergency, observation, or inpatient at a participating center. We only included patients who did not opt out of research on admission. Patients were excluded if they did not have at least one recorded ED vital sign (heart rate, respiratory rate, temperature, oxygen saturation, or systolic blood pressure) or missing comorbidity data. A complete set of vital signs was deemed necessary given our model was intended to be implemented and utilized across patients receiving a complete evaluation which would include at least one complete set of vital signs.

### Feature selection and model development

A team of subject matter experts with expertise treating patients with COVID-19 and research experience in COVID-19 identified features hypothesized to be associated with development of "severe" disease (S1 Table). To reduce the likelihood of over-fitting a Least Absolute Shrinkage and Selection Operator (LASSO)-logit model was used to facilitate feature selection from this list with the tuning parameter determined by the Bayesian information criterion (BIC) as

previously done by our group [19, 20]. LASSO is a penalized regression method that can facilitate factor selection by excluding factors with a minor contribution to the model [21]. S1 Table lists the features selected for the final model following LASSO selection.

Final features selected by LASSO included age (years), male [3, 22], race or ethnicity, non-English speaking [23, 24], overweight or obese (body mass index [BMI] > 25) [19, 25, 26], three month prior home medications [27] (defined as whether a patient was prescribed a medication within 3 months or before and after the index acute care visit) and chronic comorbidities [3, 28] extracted from ICD10 codes (S2 Table) collected in the 5 years prior to the index visit: Finally, we included the following vital signs: maximum heart rate (HR), respiratory rate (RR), temperature within the first 24 hours, and minimal peripheral arterial oxygen saturation ($SpO_2$) and systolic blood pressure (SBP) within the first 24 hours. We included in the final list of features for LASSO only the variables available on presentation to ED.

## Model construction

The purpose of this model generation was to develop a prognostic model that could predict patients who developed a severe case of COVID-19. Due to ease of interpretation and the importance to provide the basis to the clinician and patients for model predictions, a multivariable logistic regression model was trained using the features selected from LASSO. This model was developed using only data from the training dataset. A risk score was calculated in the validation cohorts based on the sum of the beta coefficients. The AUROC was calculated for all validation cohorts to evaluate discrimination in the validation datasets.

## Outcomes

Our primary outcome was "severe" COVID-19 infection, defined as intensive care unit (ICU) admission, need for invasive mechanical ventilation (ventilator use), or in-hospital or out-of-hospital mortality (defined using state death certificate database) [2, 29, 30]. The secondary outcomes were individual, and combinations of the dependent variables mentioned above.

## Training and test datasets

The training data set included 1,469 patients who were PCR-positive for SARS-CoV-2 within 14 days of an acute care, hospital-based visit including emergency department, observation, and inpatient encounters between March 4th to August 21st, 2020. The test set included 158 patients (random 90:10 selection of the training set).

## Validation datasets

We included three validation sets:

1. A temporal validation COVID-19 PCR-positive dataset comprised of 414 patients who tested positive for SARS-CoV-2 between August 22nd to October 11th, 2020. The purpose of this validation was to simulate real-time performance had the system gone "live" between August 22 and October 11th, 2020.

2. A PUI data set comprised of 13,271 patients who had a SARS-CoV-2 test with a "symptomatic" designation ordered and a result pending during the first 24 hours of an acute care, hospital-based visit irrespective of the results between May 4th and October 11th, 2020. The symptomatic designation for patients with fever, cough, dyspnea, sore throat, muscle aches, vomiting, diarrhea was based on clinical judgment and prioritized testing for faster turn-around time beginning May 4th, 2020.

3. A real-time data set included 2,174 patients with an ED visit and symptomatic test or a positive SARS-CoV-2 PCR test following implementation of the prognostic model in Emergency Departments (EDs) from November 23rd, 2020 to January 21st, 2021.

## Analysis

The patients' characteristics between data sets were compared using ANOVA and chi-square respectively for continuous versus categorical variables. Odds ratios (OR) and 95% Confidence Intervals were also reported. The sensitivity, specificity, positive predictive value (PPV), negative predictive value (NPV), likelihood ratios, false negative and false positive rate, and the area under the receiver operating characteristics (AUROC) were summarized for the model performance. Statistical significance was defined with the alpha set to 0.05, all tests were two-tailed. Statistical analyses were performed using Stata MP, version 16 (StataCorp, College Station, TX).

The real-time model was evaluated across gender and racial/ethnic groups to compare performance across different groups and ensure the model performed equitably.

## Model implementation

Implementation into an Electronic Health Record (EHR) occurred for ED patients on November 23rd, 2020. The logistic prognostic model was exported as a predictive model markup language (PMML) file. An EHR reporting workbench was developed to facilitate inputs into the model. All the inputs were mapped using corresponding ICD-10 codes (S2 Table), pharmaceutical subclasses, RxNorm codes [31], and EHR documentation flowsheets (for vitals). The output was delivered as a clinical decision support system to ED providers. For visualization purposes, the COVID-19 severity risk score was multiplied by 100 and cut points that identified patients with Low Risk (low probability of primary outcome) and High Risk (high probability of primary outcome). Visualization (S1 Fig) was highlighted on the patient sidebar, available to all ED providers and nurses, as well as physicians and staff involved in triage, patient flow, and capacity management.

## Results

### Descriptive results

A total of 2,041 patients were included in the final model training (1,469), testing (158), and temporal validation (414) (Fig 1). Table 1 listed patients' characteristics in each cohort. Overall, significant difference in all variables in demographics, use of home medications, comorbidities, and 24-hour vitals existed across training and validation cohorts, except for loop diuretic, inflammatory bowel disease, and rheumatoid arthritis. Compared to COVID-19 PCR-positive patients in the training set, the patients in the temporal validation set and PUI set were slightly younger (median age of 52.2 and 49.1 years vs. 53.6 years) and had lower rates of ICU admission (18.1% and 10.8% vs 23.4%), ventilator use (3.4% and 5.3% vs. 11.1%), and mortality (1.7% and 3.5% vs. 8.5%). Compared to the training set, the real-time data set was older (median age of 56.9 years) and had lower rates of ICU admission, ventilator use and mortality (9.4%, 3.5%, and 6.8%), respectively.

Table 2 described the odds ratios used in the logistic regression model generation. Other as race and inflammatory bowel disease, are the two variables with the highest odds ratios that reached statistical significance. Warfarin is the variable with lowest odds ratios that reached statistical significance. The model included factors that increase the odds of COVID-19 severity, such as age, male, Asian or Hispanic race, obesity, use of calcium channel blocker,

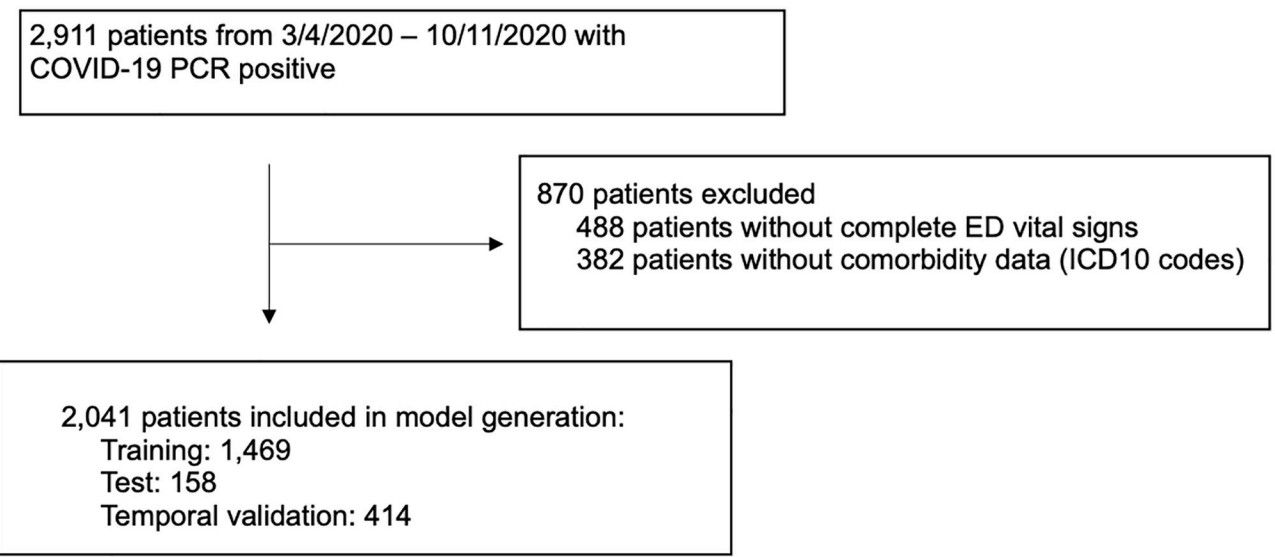

**Fig 1. Study diagram detailing the selection of patients for model generation.**

rivaroxaban, oral steroids, clopidogrel, aspirin, and a loop diuretic, hypertension, type 2 diabetes mellitus, venous thromboembolism, pacemaker/automatic implantable cardioverter-defibrillator, pulmonary hypertension, chronic kidney disease, inflammatory bowel disease, maximum temperature, heart rate, respiratory rate in 24 hours, and factors that decrease the odds, such as the use of hydrochlorothiazide, angiotensin-converting enzyme inhibitor, angiotensin II receptor blockade, warfarin, rheumatoid arthritis, minimum peripheral oxygen saturation, systolic blood pressure in 24 hours.

In the validation cohorts, the risk score was used to identify a clinically useful threshold to predict the institutional metric. Multiple thresholds were defined, and 2x2 contingency tables including sensitivity, specificity, PPV, and NPV were created for each threshold. The system leadership reviewed the various thresholds and based on clinical resources, defined an appropriate threshold. The multidisciplinary team reviewed the model performance, including sensitivity, specificity, PPV, NPV, likelihood ratios across multiple thresholds to facilitate rapid implementation. Cut-off points flagging high and low-risk patients were chosen in collaboration with both system leadership following engagement with front-line providers. The goal for low-risk cut-off was to have a high sensitivity at the expense of specificity to reduce potential errors associated with inappropriate discharge home. The goal for high-risk cut-off was a higher specificity to balance the need for close monitoring with resource scarcity, including ICU and step-down capacity.

### Pre-implementation validation: Temporal validation and in PUI

The model produced an AUROC of 0.87 (95% CI: 0.83, 0.91) for predicting the primary outcome (ICU admission, ventilator use, or death) using the temporal validation cohort (S2 Fig). None of the patients with the lowest 20% of the scores (0–0.0104) had ICU admission, ventilator use, or died, compared to 62%, 15.9%, and 7.3%, respectively, for patients with the highest 20% of the scores (0.168–1.0) (S3 Table). At a cut point of >0.1, the model had a sensitivity of 73.7% and specificity of 79.9% in predicting the composite outcomes (S4 Table).

This model was further tested in the PUI cohort that included 13,271 patients who had a SARS-CoV-2 test with a "symptomatic" designation ordered in ED. Of note, the accumulative

Table 1. Characteristics of the patients included in a training set, temporal validation set, PUI set, and real-time validation set.

| | Training Set | Test Set | Retro-Validation Set | PUI Set | Live-Validation | p-value |
|---|---|---|---|---|---|---|
| | N = 1,469 | N = 158 | N = 414 | N = 13,271 | N = 2,174 | |
| Age – median (IQR) | 53.6 (34.8–70.0) | 52.5 (35.7–67.0) | 52.2 (32.6–70.6) | 49.1 (31.3–66.8) | 56.9 (35.4–72.4) | <0.001 |
| Male – N (%) | 726 (49.4%) | 76 (48.1%) | 197 (47.6%) | 5,967 (45.0%) | 981 (45.1%) | 0.010 |
| Race – N (%) † | | | | | | <0.001 |
| White | 570 (38.8%) | 57 (36.1%) | 213 (51.4%) | 9,045 (68.2%) | 1,509 (69.4%) | |
| Black | 454 (30.9%) | 45 (28.5%) | 86 (20.8%) | 2,115 (15.9%) | 239 (11.0%) | |
| Asian | 138 (9.4%) | 22 (13.9%) | 23 (5.6%) | 1,003 (7.6%) | 145 (6.7%) | |
| Hispanic | 164 (11.2%) | 18 (11.4%) | 41 (9.9%) | 191 (1.4%) | 84 (3.9%) | |
| Declined | 121 (8.2%) | 15 (9.5%) | 38 (9.2%) | 422 (3.2%) | 163 (7.5%) | |
| Other | 22 (1.5%) | 1 (0.6%) | 13 (3.1%) | 495 (3.7%) | 34 (1.6%) | |
| Non-English Speaking – N (%) | 480 (32.7%) | 54 (34.2%) | 107 (25.8%) | 1,548 (11.7%) | 411 (18.9%) | <0.001 |
| Obesity – N (%) | 1,127 (76.7%) | 126 (79.7%) | 318 (76.8%) | 9,936 (74.9%) | 1,456 (67.0%) | <0.001 |
| Calcium Channel Blocker (home med w/in 3 months) – N (%) | 89 (6.1%) | 6 (3.8%) | 20 (4.8%) | 782 (5.9%) | 208 (9.6%) | <0.001 |
| HCTZ (home med w/in 3 months) – N (%) | 41 (2.8%) | 4 (2.5%) | 16 (3.9%) | 483 (3.6%) | 121 (5.6%) | <0.001 |
| Beta-Blocker (home med w/in 3 months) – N (%) | 172 (11.7%) | 18 (11.4%) | 50 (12.1%) | 1,786 (13.5%) | 426 (19.6%) | <0.001 |
| Ace Inhibitor (home med w/in 3 months) – N (%) | 73 (5.0%) | 4 (2.5%) | 34 (8.2%) | 952 (7.2%) | 218 (10.0%) | <0.001 |
| ARB (home med w/in 3 months) – N (%) | 91 (6.2%) | 6 (3.8%) | 23 (5.6%) | 830 (6.3%) | 193 (8.9%) | <0.001 |
| Metformin (home med w/in 3 months) – N (%) | 69 (4.7%) | 6 (3.8%) | 20 (4.8%) | 591 (4.5%) | 149 (6.9%) | <0.001 |
| Warfarin (home med w/in 3 months) – N (%) | 49 (3.3%) | 3 (1.9%) | 10 (2.4%) | 392 (3.0%) | 93 (4.3%) | 0.013 |
| Rivaroxaban (home med w/in 3 months) – N (%) | 14 (1.0%) | 3 (1.9%) | 3 (0.7%) | 209 (1.6%) | 48 (2.2%) | 0.014 |
| Oral Steroids (home med w/in 3 months) – N (%) | 86 (5.9%) | 16 (10.1%) | 18 (4.3%) | 1,087 (8.2%) | 219 (10.1%) | <0.001 |
| PPI (home med w/in 3 months) – N (%) | 217 (14.8%) | 16 (10.1%) | 61 (14.7%) | 2,303 (17.4%) | 487 (22.4%) | <0.001 |
| Clopidogrel (home med w/in 3 months) – N (%) | 20 (1.4%) | 3 (1.9%) | 9 (2.2%) | 170 (1.3%) | 61 (2.8%) | <0.001 |
| Corticosteroid Inhaler (home med w/in 3 months) – N (%) | 83 (5.7%) | 9 (5.7%) | 21 (5.1%) | 992 (7.5%) | 187 (8.6%) | 0.002 |
| Aspirin (home med w/in 3 months) – N (%) | 193 (13.1%) | 18 (11.4%) | 65 (15.7%) | 1,444 (10.9%) | 370 (17.0%) | <0.001 |
| Loop Diuretic (home med w/in 3 months) – N (%) | 104 (7.1%) | 5 (3.2%) | 21 (5.1%) | 963 (7.3%) | 181 (8.3%) | 0.17 |
| Hypertension – N (%) | 632 (43.0%) | 68 (43.0%) | 173 (41.8%) | 5,162 (38.9%) | 1,170 (53.8%) | <0.001 |
| T1DM – N (%) | 98 (6.7%) | 13 (8.2%) | 22 (5.3%) | 580 (4.4%) | 161 (7.4%) | <0.001 |
| T2DM – N (%) | 373 (25.4%) | 33 (20.9%) | 96 (23.2%) | 2,329 (17.5%) | 521 (24.0%) | <0.001 |
| Coronary Artery Disease – N (%) | 201 (13.7%) | 18 (11.4%) | 59 (14.3%) | 1,665 (12.5%) | 410 (18.9%) | <0.001 |
| VTE – N (%) | 149 (10.1%) | 10 (6.3%) | 18 (4.3%) | 928 (7.0%) | 225 (10.3%) | <0.001 |
| Heart Failure – N (%) | 176 (12.0%) | 16 (10.1%) | 39 (9.4%) | 1,413 (10.6%) | 318 (14.6%) | <0.001 |
| COPD – N (%) | 121 (8.2%) | 13 (8.2%) | 32 (7.7%) | 1,331 (10.0%) | 318 (14.6%) | <0.001 |
| Asthma – N (%) | 213 (14.5%) | 18 (11.4%) | 57 (13.8%) | 2,382 (17.9%) | 418 (19.2%) | <0.001 |
| Pacemaker/AICD – N (%) | 42 (2.9%) | 3 (1.9%) | 11 (2.7%) | 429 (3.2%) | 99 (4.6%) | 0.017 |
| Pulmonary HTN – N (%) | 74 (5.0%) | 7 (4.4%) | 12 (2.9%) | 448 (3.4%) | 128 (5.9%) | <0.001 |
| CKD – N (%) | 249 (17.0%) | 30 (19.0%) | 48 (11.6%) | 1,590 (12.0%) | 434 (20.0%) | <0.001 |
| Atrial Fib/Flutter – N (%) | 173 (11.8%) | 16 (10.1%) | 33 (8.0%) | 1,196 (9.0%) | 308 (14.2%) | <0.001 |
| CVA – N (%) | 149 (10.1%) | 16 (10.1%) | 38 (9.2%) | 1,164 (8.8%) | 298 (13.7%) | <0.001 |
| IBD – N (%) | 17 (1.2%) | 1 (0.6%) | 5 (1.2%) | 217 (1.6%) | 38 (1.7%) | 0.49 |
| Rhematoid Arthritis – N (%) | 35 (2.4%) | 2 (1.3%) | 8 (1.9%) | 317 (2.4%) | 57 (2.6%) | 0.89 |
| Malignancy – N (%) | 117 (8.0%) | 9 (5.7%) | 33 (8.0%) | 1,234 (9.3%) | 273 (12.6%) | <0.001 |
| Sleep Apnea – N (%) | 163 (11.1%) | 16 (10.1%) | 47 (11.4%) | 1,629 (12.3%) | 344 (15.8%) | <0.001 |
| Max HR in 24 hr – mean (SD) | 98.9 (20.7) | 101.2 (23.3) | 95.8 (19.5) | 95.4 (20.4) | 98.5 (21.1) | <0.001 |
| Max RR in 24 hr – mean (SD) | 25.5 (13.3) | 24.6 (10.4) | 23.4 (9.1) | 22.1 (8.4) | 23.9 (10.0) | <0.001 |
| Max Temp in 24 hr – mean (SD) | 99.7 (1.6) | 99.6 (1.7) | 99.5 (1.6) | 98.8 (1.4) | 99.0 (1.4) | <0.001 |

(Continued)

**Table 1.** (Continued)

| | Training Set | Test Set | Retro-Validation Set | PUI Set | Live-Validation | p-value |
|---|---|---|---|---|---|---|
| | **N = 1,469** | **N = 158** | **N = 414** | **N = 13,271** | **N = 2,174** | |
| Min SpO2 in 24 hr – mean (SD) | 92.1 (8.2) | 91.8 (8.1) | 93.1 (6.1) | 94.7 (5.4) | 92.8 (6.7) | <0.001 |
| Min SPB in 24 hr – mean (SD) | 112.1 (22.7) | 110.5 (20.7) | 114.9 (21.2) | 122.7 (20.6) | 116.6 (21.3) | <0.001 |
| ICU Admission – N (%) | 346 (23.6%) | 34 (21.5%) | 75 (18.1%) | 1,428 (10.8%) | 100 (9.4%) | <0.001 |
| Mechanical Ventilation – N (%) | 164 (11.2%) | 17 (10.8%) | 14 (3.4%) | 478 (5.3%) | 37 (3.5%) | <0.001 |
| Died – N (%) | 125 (8.5%) | 14 (8.9%) | 7 (1.7%) | 460 (3.5%) | 73 (6.8%) | <0.001 |
| Bad Outcome* – N (%) | 382 (26.0%) | 38 (24.1%) | 76 (18.4%) | 1,627 (12.3%) | 247 (11.4%) | <0.001 |

*Primary outcome is defined as ICU admission, need for mechanical ventilation, or death.

Continuous, normally distributed variables (mean and SD will be reported) were compared using ANOVA.

Continuous, non-normally distributed variables (median and IQR reported) were compared using Wilcoxon rank-sum (2 groups) or Kruskal-Wallis (>2 groups) test.

Categorical and binary groups were compared using Pearson's chi-squared test.

**Abbreviations**: HCTZ: hydrochlorothiazide; ACE: angiotensin-converting enzyme; ARB: angiotensin receptor blocker; PPI: proton pump inhibitor; T1DM: Type 1 diabetes mellitus; T2DM: Type 2 diabetes mellitus; VTE: venous thromboembolism; COPD: chronic obstructive pulmonary disease; AICD: automatic implantable cardioverter-defibrillator; HTN: hypertension; CKD: chronic kidney disease; CVA: cerebrovascular accident; Afib: atrial fibrillation; Aflutter: Atrial flutter; IBD: inflammatory bowel disease; hr: hour; Max: maximum; Min: minimum; HR: heart rate; RR: respiratory rate; Temp: temperature; SpO$_2$: peripheral oxygen saturation; SBP: systolic blood pressure; ICU: intensive care unit.

COVID positive rate in the PUI data set was 26.8% (3,561 of 13,271 patients). A total of 68% of patients were discharged before the test resulted in our medical system. The model produced an AUROC of 0.82 (95% CI: 0.81, 0.83) for predicting the composite outcomes in the PUI cohort (S3 Fig). For patients with the lowest 20% of the scores (0.00062–0.0074), only 1.0% had ICU admission, 0.3% ventilator use, and 0.2% died, compared to 31.6%, 13.8%, and 11.9%, respectively, for patients with the highest 20% of the scores (0.168–1.0) (S5 Table). At the cut point of >0.1, the model had a sensitivity of 52.2% and specificity of 88.1% in predicting composite outcomes (S5 Table).

## Real-time validation

Critically, we implemented this model to predict the composite outcomes to evaluating the COVID-19 severity and assessed the model's real-time performance. The COVID positive rate in the real-time validation set was 61.2% (1,331 of 2,174 patients). This real-time cohort had a median age of 56.9 years (IQR: 35.4–72.4), had an ICU admission rate of 9.4%, ventilation rate of 3.5%, and mortality rate of 6.8%. The model had an AUROC of 0.85 (95% CI, 0.83, 0.87) to predict the primary outcome in the real-time data set. (Fig 2). The rates of ICU admission, ventilator use, and death in patients with the lowest 20% of the scores (0.001–0.009) were zero, significantly lower compared to those rates (32.7%, 15.5%, and 22.0%, respectively) for patients with the highest 20% of the scores (0.20–0.99) (Table 3). At the cut point of >0.1, the model had a sensitivity of 78% and a specificity of 71% in the real-time data set (Table 4). To evaluate the probabilities in the real time world we depicted the calibration plot for the real-time validation set. (S4 Fig).

## Model performance on individual and combined outcomes and across minorities

The AUROC of all cohorts predicting various outcomes combined and individual are listed in Table 5. The performance remained strong for predicting secondary outcomes in combinations of ICU admission, need for mechanical ventilation, and mortality.

**Table 2. Odds ratios of variables in the model.**

| Variables | Odds Ratio | 95% CI | p-value |
|---|---|---|---|
| Age | 1.03 | 1.02, 1.04 | <0.001 |
| Male | 1.95 | 1.33, 2.84 | 0.001 |
| Race | | | |
| Black | 0.94 | 0.54, 1.62 | 0.817 |
| Asian | 1.59 | 0.77, 3.26 | 0.208 |
| Hispanic | 1.49 | 0.67, 3.30 | 0.329 |
| Declined | 1.36 | 0.62, 2.97 | 0.437 |
| Other | 4.13 | 1.39, 12.33 | 0.011 |
| Non-English speaking | 1.17 | 0.69, 1.99 | 0.560 |
| Obesity | 1.41 | 0.89, 2.22 | 0.140 |
| Calcium channel blocker (home med w/in 3 months) | 1.37 | 0.67, 2.76 | 0.386 |
| HCTZ (home med w/in 3 months) | 0.60 | 0.21, 1.69 | 0.330 |
| Beta-blocker (home med w/in 3 months) | 1.00 | 0.54, 1.84 | 0.991 |
| ACE inhibitor (home med w/in 3 months) | 0.49 | 0.20, 1.18 | 0.112 |
| ARB (home med w/in 3 months) | 0.83 | 0.40, 1.71 | 0.606 |
| Metformin (home med w/in 3 months) | 0.86 | 0.38, 1.98 | 0.727 |
| Warfarin (home med w/in 3 months) | 0.25 | 0.10, 0.68 | 0.006 |
| Rivaroxaban (home med w/in 3 months) | 1.34 | 0.38, 4.69 | 0.644 |
| Oral steroids (home med w/in 3 months) | 1.45 | 0.69, 3.03 | 0.328 |
| PPI (home med w/in 3 months) | 0.99 | 0.60, 1.63 | 0.970 |
| Clopidogrel (home med w/in 3 months) | 1.95 | 0.60, 6.32 | 0.266 |
| Corticosteroid inhaler (home med w/in 3 months) | 1.12 | 0.53, 2.38 | 0.769 |
| Aspirin (home med w/in 3 months) | 1.39 | 0.82, 2.34 | 0.217 |
| Loop diuretic (home med w/in 3 months) | 1.58 | 0.85, 2.93 | 0.150 |
| Hypertension | 1.44 | 0.91, 2.27 | 0.119 |
| T1DM | 0.83 | 0.40, 1.72 | 0.620 |
| T2DM | 1.44 | 0.93, 2.23 | 0.103 |
| Coronary artery disease | 0.66 | 0.39, 1.12 | 0.125 |
| VTE | 1.37 | 0.81, 2.34 | 0.244 |
| Heart failure | 1.01 | 0.57, 1.79 | 0.965 |
| COPD | 0.96 | 0.53, 1.75 | 0.896 |
| Asthma | 1.00 | 0.58, 1.74 | 0.992 |
| Pacemaker/AICD | 2.07 | 0.90, 4.79 | 0.088 |
| Pulmonary HTN | 1.44 | 0.70, 2.96 | 0.323 |
| CKD | 1.62 | 0.99, 2.66 | 0.056 |
| Afib/Aflutter | 1.04 | 0.60, 2.78 | 0.897 |
| CVA | 1.27 | 0.75, 2.16 | 0.370 |
| IBD | 3.95 | 1.10, 14.23 | 0.036 |
| Rheumatoid arthritis | 0.62 | 0.23, 1.72 | 0.363 |
| Malignancy | 1.03 | 0.57, 1.87 | 0.928 |
| Sleep apnea | 0.87 | 0.51, 1.49 | 0.616 |
| Max HR in 24 hr | 1.01 | 1.00, 1.02 | 0.275 |
| Max RR in 24 hr | 1.02 | 1.01, 1.03 | 0.001 |
| Max Temp in 24 hr | 1.34 | 1.20, 1.51 | <0.001 |
| Min SpO2 in 24 hr | 0.94 | 0.92, 0.96 | <0.001 |

*(Continued)*

**Table 2.** (Continued)

| Variables | Odds Ratio | 95% CI | p-value |
|---|---|---|---|
| Min SPB in 24 hr | 0.98 | 0.97, 0.99 | <0.001 |

**Abbreviations**: CI: confidence interval; w/in: within; HCTZ: hydrochlorothiazide; ACE: angiotensin-converting enzyme; ARB: angiotensin receptor blocker; PPI: proton pump inhibitor; T1DM: Type 1 diabetes mellitus; T2DM: Type 2 diabetes mellitus; VTE: venous thromboembolism; COPD: chronic obstructive pulmonary disease; AICD: automatic implantable cardioverter-defibrillator; HTN: hypertension; CKD: chronic kidney disease; CVA: cerebrovascular accident; Afib: atrial fibrillation, Aflutter: atrial flutter, IBD: inflammatory bowel disease; hr: hour; Max: maximum; Min: minimum; HR: heart rate; RR: respiratory rate; Temp: temperature; $SpO_2$: peripheral oxygen saturation; SBP: systolic blood pressure.

Furthermore, the model performed equitably across gender/racial/ethnic minorities (Table 6). The AUROC for Blacks and Asians are 0.94 (95% CI: 0.82.1.0) and 0.94 (95% CI: 0.90. 0.99), compared to that for Whites 0.82 (95% CI: 0.78, 0.86) (p>0.05) (Table 6). There is no statistical difference between the model performance in female versus male patients (p>0.05).

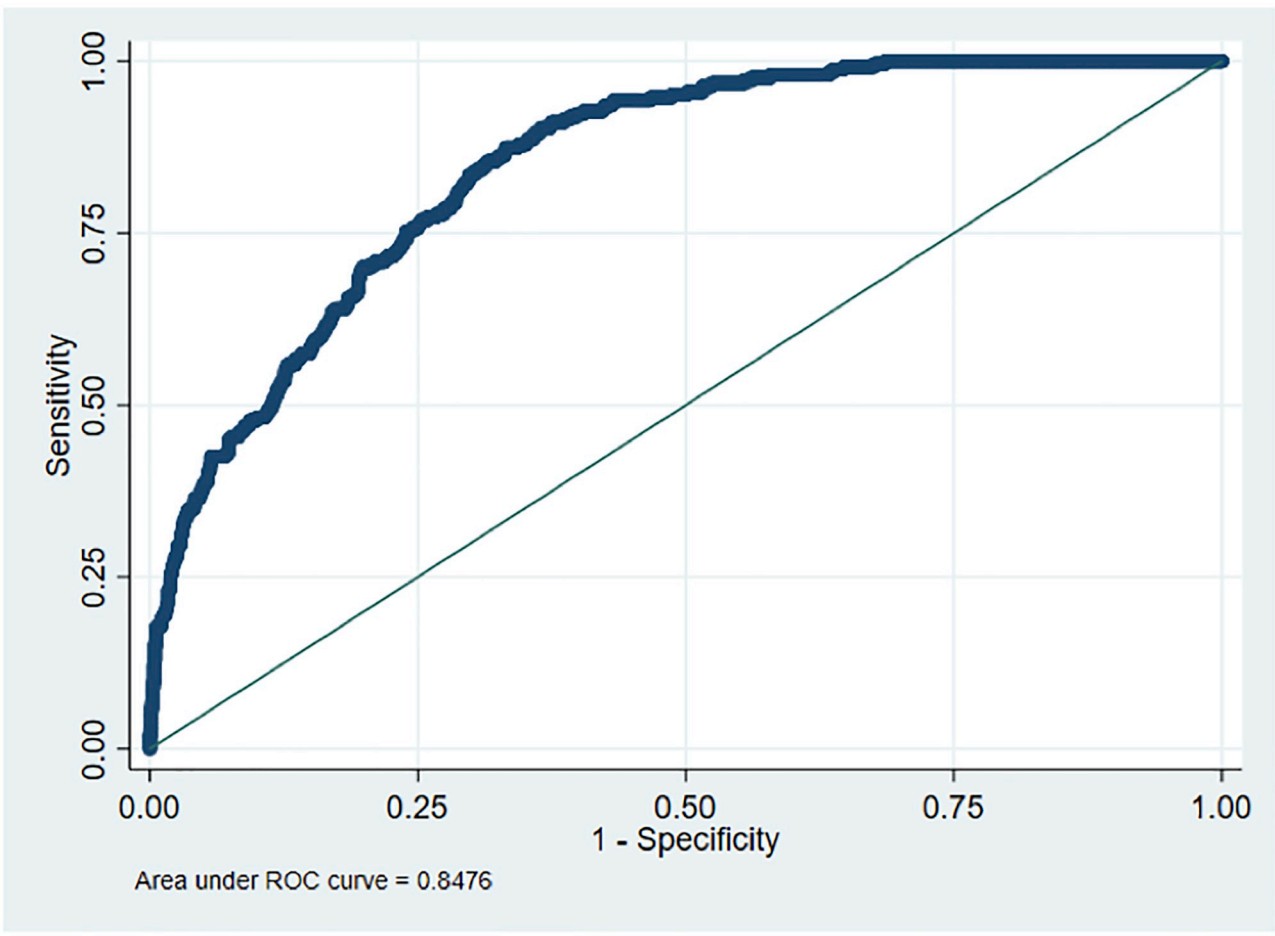

**Fig 2. ROC curve for real-time validation (n = 2,174).**

**Table 3. Distribution of outcomes by score ranges in quintile for the real-time validation data set (n = 2,174).**

|  | Score Range | ICU, n (%) | Vent, n (%) | Death, n (%) | Primary outcome (%) | n |
|---|---|---|---|---|---|---|
| **Lowest 20% scores** | 0.001–0.009 | 0 (0.0%) | 0 (0.0%) | 0 (0.0%) | 0 (0.0%) | 435 |
| **20–40%** | 0.009–0.02 | 8 (4.1%) | 5 (2.6%) | 1 (0.5%) | 8 (1.8%) | 435 |
| **40–60%** | 0.02–0.06 | 21 (10.1%) | 5 (2.5%) | 4 (2.0%) | 23 (5.3%) | 435 |
| **60–80%** | 0.06–0.20 | 59 (22.2%) | 14 (5.6%) | 21 (8.4%) | 72 (16.6%) | 435 |
| **Highest 20% of scores** | 0.20–0.99 | 105 (32.7%) | 46 (15.5%) | 65 (22.0%) | 144 (33.2%) | 434 |

**Abbreviations**: ICU: Intense Care Unit; Vent: Ventilator.

**Table 4. Clinical performance of the logistical model for predicting COVID-19 severity\* in the real-time validation data set (n = 2174).**

| Cut point | True + | False + | True - | False - | Sensitivity | Specificity | NPV | PPV | LR + | LR - |
|---|---|---|---|---|---|---|---|---|---|---|
| >0.03 | 238 | 1032 | 983 | 12 | 95% | 49% | 0.19 | 0.99 | 1.86 | 0.10 |
| >0.05 | 228 | 828 | 1187 | 22 | 91% | 59% | 0.22 | 0.98 | 2.22 | 0.15 |
| >0.07 | 214 | 699 | 1316 | 36 | 86% | 65% | 0.23 | 0.97 | 2.47 | 0.22 |
| >0.09 | 197 | 618 | 1397 | 53 | 79% | 69% | 0.24 | 0.96 | 2.57 | 0.31 |
| >0.1 | 194 | 587 | 1428 | 56 | 78% | 71% | 0.25 | 0.96 | 2.66 | 0.32 |
| >0.11 | 189 | 553 | 1462 | 61 | 76% | 73% | 0.25 | 0.96 | 2.75 | 0.34 |
| >0.13 | 178 | 495 | 1520 | 72 | 71% | 75% | 0.26 | 0.95 | 2.90 | 0.38 |
| >0.15 | 166 | 459 | 1556 | 84 | 66% | 77% | 0.27 | 0.95 | 2.91 | 0.44 |
| >0.17 | 158 | 415 | 1600 | 92 | 63% | 79% | 0.28 | 0.95 | 3.07 | 0.46 |
| >0.19 | 150 | 387 | 1628 | 100 | 60% | 81% | 0.28 | 0.94 | 3.12 | 0.50 |

\*COVID-19 severity is defined as ICU admission, need for mechanical ventilation, or death.

## Discussion

We developed and implemented a ML-enabled model to predict increased risk for COVID-19 severity to support the ED physicians' clinical decision-making across our 12-sites medical system. Despite the significant variabilities of the factors, our model performed well in a large PUI study population. This approach is beneficial for clinical decision-making in ED where the COVID-19 PCR test results are inconsistently resulted. Importantly, we evaluated our model real-time in PUI patients seeking acute care in ED after the score became available in the EHR and the model performance remained strong. The difference in ICU admission rate, ventilator use, and mortality rate between the training set and the temporal, PUI, and real-time validation sets can be explained by the temporal improvement in COVID-19 patients' outcomes that was noted in other studies [4, 32, 33]. The COVID-19 ICU admission and patient survival improved in our study over time as it did in other reports, perhaps because of

**Table 5. AUROC of all cohorts for predicting the individual and composite outcomes.**

| Cohort | ICU + Vent + Death outcome | Vent + Death outcome | ICU outcome | Vent outcome | Death outcome |
|---|---|---|---|---|---|
| Test set | 0.86 | 0.88 | 0.84 | 0.86 | 0.86 |
| Temporal validation set | 0.87 | 0.93 | 0.87 | 0.93 | 0.91 |
| PUI set | 0.82 | 0.85 | 0.82 | 0.80 | 0.85 |
| Real-time set | 0.85 | 0.86 | 0.78 | 0.79 | 0.83 |

**Abbreviations**: ICU: intensive care unit; Vent: ventilator.

**Table 6. Sensitivity analysis across gender/racial/ethnic minorities.**

| Variable | AUROC | 95% CI* |
|---|---|---|
| Female | 0.88 | 0.85–0.92 |
| Male | 0.84 | 0.79–0.89 |
| White | 0.82 | 0.78–0.86 |
| Black | 0.94 | 0.82–1.0 |
| Asian | 0.94 | 0.90–0.99 |
| Hispanic | 0.96 | 0.91–1.0 |
| Declined | 0.85 | 0.77–0.92 |
| Other | 0.97 | 0.91–1.0 |

* All p-values were > 0.05.

better understating of the diseases and improvement of treatment as the pandemic progresses [4, 33] In our study, for a cutoff of 0.1 for COVID-19 severity, our model had a sensitivity of 73.7% and specificity 79.9% in the prospective validation set, 52.2% and 88.1%, respectively in the PUI set, and 78% and 71%, respectively in the real-time validation set. These results show good discrimination for patients with scores associated with increased rates of the primary outcome. Furthermore, the performance of the models were robust to the secular improvements in outcomes throughout validation.

Our model purpose was to estimate the risk of severe disease using ML as CDS in patients with or suspicion of COVID-19 presenting in ED. Furthermore, our goal was to use this model as CDS and facilitate the shared decision making between ED providers and patients regarding ED discharge and home saturation monitoring. The variables included (demographics, comorbidities, home medication, vital signs) are readily available in the ED. The laboratory values which are not always obtainable in ED were not included in the final model, which seemed to be feasible as described in a recent ML model published in the literature [34]. The variables associated with a significantly higher risk for COVID-19 severity in our model were male gender, older age, other as race, increased temperature, increased respiratory rate, decreased oxygen saturation, inflammatory bowel disease. Comparable to our model, vital signs, age, BMI, and comorbidities were the most important predictors in other investigations and reviews [35, 36]. Oxygen saturation and patient's age were strong risk factors for deterioration and mortality in COVID-19 in a systematic evaluation of predictive models [13]. The use of warfarin appeared to be protective for our study's composite outcome, similar to another report [37]. Hypercoagulability and need for anticoagulation were well recognized in COVID-19 and likely from increased immune response [38, 39]. We included variables that were not significant on univariate analysis as well as variables that were protective. These variables made our model valuable in real life when many covariates and confounding factors exist and increased the model calibration.

It is imperative that ML models are evaluated for equity across gender, race and ethnicity. We included gender, race and ethnicity in our model given the association between minority populations and male gender and worse COVID-19 outcomes [23, 24, 40–42]. While others chose to create a different prognostic model for males and females, we decided to include all [18]. The male gender was a significant predictor in our study, and the AUROC in male patients showed good performance without statistical difference compared to the female gender AUROC. While including race has led to over and undertreatment of minority populations [43, 44], due to sampling bias, others argue that creating a "race un-aware" model also pertains risk in specific situations [45]. One particular situation is when race/ethnicity is associated with increased risk of the outcome, like other as race in our study that showed increase

risk of COVID-19 severity. By creating a model without race or ethnicity, the model is trained to reflect the majority population and will inherently underappreciate the risk across minority populations [45]. Our model performed equitably across racial/ethnic minorities and did not increase the risk of widening the disparate outcomes observed throughout the pandemic [24, 46, 47]. By increasing treatment and resource allocation to non-whites, we hypothesize that this will increase equitable treatment allocation and attenuate disparate care.

Unlike most prognostic models predicting the COVID-19 diagnosis [35, 48–50], our study aimed to implement and assess the predictive model in patients with suspected COVID-19 disease, or PUI. It is worth noting that 68% of our patients were discharged before the test resulted in our medical system. During this uncertainty period, many ED physicians are required to make clinical and triage decisions. Previous predictive models for patients with suspected COVID-19 infection have used imaging, demographics, signs and symptoms, vital signs to predict the likelihood of COVID-19 diagnosis, but they have not sought to predict the severity of the disease [12, 14, 51]. The Epic Deterioration Index (EDI) is a proprietary emergency deterioration index that has been developed in 3 US hospitals in US between 2012 and 2016; although it is not specific for COVID-1, it has been introduced in over 100 US hospitals to predict COVID-19 deterioration [30].

Multiple prognostic models for COVID-19 have been previously developed [14, 18, 30, 52, 53]. However, previous models suffer from multiple limitations. For example, many prior prognostic models included very limited training dataset [54, 55]. The largest study to date published in Great Britain used the 4 C Mortality Score to stratify the severity of the COVID-19 [56]. In contrast to our model, the 4 C includes some laboratory values (urea level and C-reactive protein) not always available in ED, and used data from COVID-19 positive patients admitted to the hospital: AUROC 0.79, (95% CI 0.78–0.79). A systematic external validation of 22 prognostic models in a cohort of 411 patients with COVID-19 found that NEWS2 score that predicted ICU admission or death within 14 days for symptoms onset: AUROC 0.78 (95% CI 0.73–0.83) achieved the highest AUROC [13]. The EDI has been recently tested on 392 COVID-19 hospitalized patients in single center and found an AUROC 0.79 (95% CI, 0.74–0.84) [30]. Our model performance for predicting COVID-19 severity in our prospective validation, PUI, and real-time data sets is more robust than in the above mentioned external validation of the prognostic models. Data from the national Registry of suspected COVID-19 in Emergency care (RECOVER network) comprising 116 hospitals from 25 states in the US produced a 13 variable score that can predict the probability of infection in patients presenting with suspected COVID-19 in ED [57]. The large RECOVER registry used patient data such as age, temperature, oxygen saturation, symptoms, and ethnicity readily available in ED; however, the score was developed with retrospective data and it was not tested in real time [57].

## Strengths and limitations

Our study has several strengths. *First*, it was validated on patients with COVID-19 diagnosis and patients with suspected COVID-19. *Second*, the logistic regression-based ML used data readily available in ED. *Third*, we included variables that were non-significant or were protective in univariate analysis, making the logistic regression-based ML more suitable for real-life when many confounders exist. *Fourth*, it was tested in real-time in patients with suspected COVID-19 who presented in the acute care setting as a CDS for ED providers and patients. *Finally*, our model was tested for gender and race/ethnicity differences and performed equitably to avoid disparities.

These findings must be viewed within the context of the following limitations. *First*, this study was done within a single healthcare system. Despite a large catchment area that includes

surrounding states, these results are specific to the regional patient population in which the models were derived until they have been validated in other populations with different demographics and socioeconomic backgrounds. *Second*, our model over-predicted the disease severity making it a valuable tool for patient safety and less for resource utilization. *Third*, the accuracy of patient comorbidities and medications available in ED relies on the history from EHR, not consistently updated during the acute care visit. *Fourth*, as seen in the calibration plot, the model does suffer from at the high-risk end, this is likely due to imbalance of the dataset without a large degree of "bad outcomes". Future studies will seek to increase sample size and further include external institutions which will aid in further optimization of the model along with addressing the generalizability, respectively. *Lastly*, this study sought to develop, validate, and implement a prediction model to support clinical decision-making. Importantly, the model was never intended to replace clinical judgment, rather it was intended to complement and better inform providers and patients, specifically when there is a large degree of clinical uncertainty. The effect on clinical decisions and the long-term effect on patient safety remained to be determined and were beyond the scope of this analysis.

## Conclusions

COVID-19 has burdened healthcare systems from multiple different facets, and finding ways to alleviate stress is crucial. CDS through ML-enabled predictive modeling may add to patient care, reduce undue decision-making variations, and optimize resource utilization, especially during a pandemic. We present a 12-hospital successful development and implementation of a COVID-19 prediction model that performs well across gender, race, and ethnicity for three different outcomes. The severity of illness primary outcome performed well in the PUI population despite being developed on a COVID-19 positive population. The effect on patient outcomes and resource use are needed to assess further the benefits of the model presented here.

## Supporting information

**S1 Table. Factors of interest and factors selected for the final model.**
(DOCX)

**S2 Table. Categories of comorbidities and ICD 10 Codes.**
(DOCX)

**S3 Table. Distribution of outcomes by score ranges in quintile for the temporal validation data set (n = 414).**
(DOCX)

**S4 Table. Clinical performance of the logistical model for predicting COVID-19 disease severity\* in the temporal validation data set (n = 414).**
(DOCX)

**S5 Table. Distribution of outcomes by score ranges in quintile for the PUI data set (n = 13,271).**
(DOCX)

**S6 Table. Clinical performance of the logistical model for predicting COVID-19 severity\* in the PUI data set (n = 13,271).**
(DOCX)

**S1 Fig. Implementation of the model for predicting COVID-19 severity in ED.**
(PDF)

**S2 Fig. ROC curve for prospective validation (n = 414).**
(PDF)

**S3 Fig. ROC curve for PUI validation (n = 13,271).**
(PDF)

**S4 Fig. Real-time validation calibration plot.**
(PDF)

## Acknowledgments

The authors wish to thank Sean Switzer, DO, Eric Murray, and Iva Ninkovic for valuable technical support, and Gyorgy Simon, PhD, for regression analysis and model building advice.

## Author Contributions

**Conceptualization:** Monica I. Lupei, Danni Li, Nicholas E. Ingraham, Karyn D. Baum, Bradley Benson, Michael Puskarich, David Milbrandt, Genevieve B. Melton, Daren Scheppmann, Michael G. Usher, Christopher J. Tignanelli.

**Data curation:** Monica I. Lupei, Danni Li, Nicholas E. Ingraham, Karyn D. Baum, Daren Scheppmann, Michael G. Usher, Christopher J. Tignanelli.

**Formal analysis:** Monica I. Lupei, Danni Li, Nicholas E. Ingraham, Michael G. Usher, Christopher J. Tignanelli.

**Investigation:** Monica I. Lupei, Nicholas E. Ingraham, Genevieve B. Melton, Daren Scheppmann, Michael G. Usher, Christopher J. Tignanelli.

**Methodology:** Monica I. Lupei, Danni Li, Nicholas E. Ingraham, Karyn D. Baum, David Milbrandt, Genevieve B. Melton, Michael G. Usher, Christopher J. Tignanelli.

**Project administration:** Monica I. Lupei, Genevieve B. Melton.

**Resources:** Nicholas E. Ingraham, Karyn D. Baum, Michael G. Usher.

**Software:** Michael G. Usher, Christopher J. Tignanelli.

**Supervision:** Monica I. Lupei, Genevieve B. Melton, Christopher J. Tignanelli.

**Validation:** Christopher J. Tignanelli.

**Visualization:** Monica I. Lupei, Danni Li, Michael G. Usher.

**Writing – original draft:** Monica I. Lupei, Danni Li, Nicholas E. Ingraham, Michael Puskarich, Michael G. Usher, Christopher J. Tignanelli.

**Writing – review & editing:** Monica I. Lupei, Danni Li, Nicholas E. Ingraham, Karyn D. Baum, Bradley Benson, Michael Puskarich, David Milbrandt, Genevieve B. Melton, Michael G. Usher, Christopher J. Tignanelli.

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
