## [Decision Letter · Decision Letter 0]

1 Oct 2021

PONE-D-21-26126A 12-hospital prospective evaluation of a clinical decision support prognostic algorithm based on logistic regression as a form of machine learning to facilitate decision making for patients with suspected COVID-19PLOS ONE

Dear Dr. Lupei,

Thank you for submitting your manuscript to PLOS ONE. After careful consideration, we feel that it has merit but does not fully meet PLOS ONE’s publication criteria as it currently stands. Therefore, we invite you to submit a revised version of the manuscript that addresses the points raised during the review process.

We look forward to receiving your revised manuscript.

Kind regards,

Alpamys Issanov

Academic Editor

PLOS ONE

Journal Requirements:

2. Please note that PLOS ONE has specific guidelines on code sharing for submissions in which author-generated code underpins the findings in the manuscript. In these cases, all author-generated code must be made available without restrictions upon publication of the work. Please review our guidelines at https://journals.plos.org/plosone/s/materials-and-software-sharing#loc-sharing-code and ensure that your code is shared in a way that follows best practice and facilitates reproducibility and reuse

Additional Editor Comments (if provided):

1) Please include all information in Financial Disclosure section as per the PLOS ONE submission guidelines.

2) All abbreviations in first mention in abstract and introduction must be spelled out.

3) The manuscript needs English proof-editing (several typos, misspellings, incomplete sentences, etc)

4) The COVID-19 statistics in the introduction need to be updated.

Reviewers' comments:

Reviewer's Responses to Questions

**Comments to the Author**

1. Is the manuscript technically sound, and do the data support the conclusions?

Reviewer #1: Yes

Reviewer #2: Partly

2. Has the statistical analysis been performed appropriately and rigorously? 

Reviewer #1: Yes

Reviewer #2: Yes

3. Have the authors made all data underlying the findings in their manuscript fully available?

Reviewer #1: Yes

Reviewer #2: Yes

4. Is the manuscript presented in an intelligible fashion and written in standard English?

Reviewer #1: Yes

Reviewer #2: Yes

5. Review Comments to the Author

Reviewer #1: The Authors have designed a model for the severity assessment of COVID-19 patients. The proposed model is simple, efficient and shows good predictive performance. The model has been validated over a very large set of subjects which proves the applicability of the model. The in-depth analysis of the model's performance also gives confidence in the model.

Comments:

1. Test set n=158 statistics not reported in table 1.

2. In validation sets B and C, all subjects considered may not be COVID-19 positive (as all patients’ RT-PCR test outcome is unknown). Statistics of how many patients had a positive RT-PCR result must be reported. Further, the severity rate in cohorts B, C are considerably lower than in cohort A.

3. What criteria were considered for the final list of features to be selected after lasso feature selection?

4. The cut-off points for balanced sensitivity and specificity are around 0.1, which means that the model predictions do not signify real probabilities of severe predictions, limiting the use of the predicted score in the real world. The authors can look into model/probability calibration to fix this.

Reviewer #2: This is an interesting manuscript which has obtained a robust COVID-19 prediction model that performs well across gender, race, and ethnicity for three different outcomes. The main problem with this manuscript is that the expected benefit of a CDSS to alleviate burdened and stressed healthcare systems is not demonstrated. This manuscript describes the construction of the model and its predictive capacity, but it has not assessed the impact this model have had on supporting clinical decision making, patient outcomes, or resource use. Authors themselves indicate the need to assess further the benefits of the model presented in this manuscript.

6. PLOS authors have the option to publish the peer review history of their article (what does this mean?). If published, this will include your full peer review and any attached files.

Reviewer #1: No

Reviewer #2: No

---

## [Author Response · Author response to Decision Letter 0]

19 Nov 2021

November 15th, 2021

Alpamys Issanov

Academic Editor

PLOS ONE

Dear Dr. Issanov,

Thank you very much for considering our manuscript PONE-D-21-26126

“A 12-hospital prospective evaluation of a clinical decision support prognostic algorithm based on logistic regression as a form of machine learning to facilitate decision making for patients with suspected COVID-19” for publication in PLOS One after revisions.

We appreciate the editor and reviewers’ comments and suggestions, and we addressed them by revising and improving our manuscript. We responded to each reviewers’ comment in our detailed Response to Reviewers letter. We also added the Final Disclosure Statement in this resubmission. 

Our manuscript brings our experience developing and implementing the prognostic model for COVID-19 severity in real-time in the emergency department to the PLOS One readers. We certify that our manuscript is original and has not been published elsewhere. We look forward to receiving your response. Please address all correspondence related to this manuscript to lupei001@umn.edu.

Sincerely,

Monica Lupei

---

## [Decision Letter · Decision Letter 1]

20 Dec 2021

A 12-hospital prospective evaluation of a clinical decision support prognostic algorithm based on logistic regression as a form of machine learning to facilitate decision making for patients with suspected COVID-19

PONE-D-21-26126R1

Dear Dr. Lupei,

We’re pleased to inform you that your manuscript has been judged scientifically suitable for publication and will be formally accepted for publication once it meets all outstanding technical requirements.

Kind regards,

Alpamys Issanov

Academic Editor

PLOS ONE

Additional Editor Comments (optional):

Reviewers' comments:

Reviewer's Responses to Questions

**Comments to the Author**

1. If the authors have adequately addressed your comments raised in a previous round of review and you feel that this manuscript is now acceptable for publication, you may indicate that here to bypass the “Comments to the Author” section, enter your conflict of interest statement in the “Confidential to Editor” section, and submit your "Accept" recommendation.

Reviewer #1: (No Response)

Reviewer #2: (No Response)

2. Is the manuscript technically sound, and do the data support the conclusions?

Reviewer #1: (No Response)

Reviewer #2: Yes

3. Has the statistical analysis been performed appropriately and rigorously? 

Reviewer #1: (No Response)

Reviewer #2: Yes

4. Have the authors made all data underlying the findings in their manuscript fully available?

Reviewer #1: (No Response)

Reviewer #2: Yes

5. Is the manuscript presented in an intelligible fashion and written in standard English?

Reviewer #1: (No Response)

Reviewer #2: Yes

6. Review Comments to the Author

Reviewer #1: (No Response)

Reviewer #2: Authors have responded to the concerns indicated. The manuscript has improved and it could be accepted for publication in this updated version.

7. PLOS authors have the option to publish the peer review history of their article (what does this mean?). If published, this will include your full peer review and any attached files.

Reviewer #1: No

Reviewer #2: No

---

## [Editor Report · Acceptance letter]

26 Dec 2021

PONE-D-21-26126R1 

A 12-hospital prospective evaluation of a clinical decision support prognostic algorithm based on logistic regression as a form of machine learning to facilitate decision making for patients with suspected COVID-19 

Dear Dr. Lupei:

I'm pleased to inform you that your manuscript has been deemed suitable for publication in PLOS ONE. Congratulations! Your manuscript is now with our production department. 

Kind regards, 

on behalf of

Dr. Alpamys Issanov 

Academic Editor

PLOS ONE